# Clinical Relevance of *TP53* Mutation and Its Characteristics in Breast Cancer with Long-Term Follow-Up Date

**DOI:** 10.3390/cancers16233899

**Published:** 2024-11-21

**Authors:** Seung Hyun Hwang, Seung Ho Baek, Min Ji Lee, Yoonwon Kook, Soong June Bae, Sung Gwe Ahn, Joon Jeong

**Affiliations:** 1Institute for Breast Cancer Precision Medicine, Yonsei University College of Medicine, Seoul 06273, Republic of Korea; kelvin9999@hanmail.net (S.H.H.); holydante@yuhs.ac (S.H.B.); skymindor@yuhs.ac (M.J.L.); yoonwon7@yuhs.ac (Y.K.); mission815815@yuhs.ac (S.J.B.); asg2004@yuhs.ac (S.G.A.); 2Department of Breast and Thyroid Surgery, Sam Hospital, Anyang 14030, Republic of Korea; 3Department of Surgery, Gangnam Severance Hospital, Yonsei University College of Medicine, Seoul 06273, Republic of Korea

**Keywords:** *TP53* mutation, missense mutation, missense hotspot, breast cancer, recurrence-free survival, overall survival

## Abstract

The *TP53* mutation is one of the prevalent genetic alterations in human cancers and is often linked to a poor prognosis. While earlier studies have produced mixed results, they frequently involved small patient groups focused on specific breast cancer subtypes and treatments. To clarify these findings, we examined the clinical relevance of *TP53* mutations in 650 patients across all subtypes, with consistent treatment based on subtype. In total, 172 (26.5%) had *TP53* mutations, including 34 (19.8%) with missense hotspot mutations. Those with *TP53* mutations had worse outcomes, with a 10-year recurrence-free survival rate of 83.5% compared to 86.6% for those without (*p* = 0.026), and a 10-year overall survival rate of 88.1% versus 91.0% (*p* = 0.003). However, the outcomes among patients with *TP53* mutation did not differ significantly by mutation types or locations. Consequently, further research is necessary to explore the clinical relevance of the characteristics of *TP53* mutation.

## 1. Introduction

The *TP53* gene, which codes for the tumor-suppressor protein p53, is the most frequently mutated gene in human cancers [1]. Located on chromosome 17p13.1, *TP53* consists of 11 exons, 10 introns, and 393 amino acid residues, and encodes the p53 protein, a transcription factor with distinct amino-terminal, DNA-binding, and carboxy-terminal domains [2]. The *TP53*-activated pathway exerts tumor-suppressive functions by regulating DNA repair, cell-cycle arrest, senescence, and apoptosis, thereby inhibiting early tumorigenesis, tumor growth, and progression [3,4,5]. As a result, the activation of p53 in normal tissues is critical for preventing tumorigenesis. However, tumors with *TP53* mutations not only lose these tumor-suppressive functions but also often acquire gain-of-function mutations that promote tumor growth [6,7]. Consequently, *TP53*-mutated tumors typically exhibit rapid progression, resistance to treatment, and a poor prognosis [8,9,10].

According to the International Agency for Research on Cancer (IACR) database, over 75% of *TP53* mutations were missense mutations, with approximately 97% located in exons encoding the DNA-binding domain (DBD, residue 98-292). Six codons (175, 220, 245, 248, 273, and 282) are recognized as well-known missense hotspots, each accounting for more than 2% of all missense mutations (https://www.cbioportal.org/, (accessed on 29 October 2024)). These single nucleotide substitutions disrupt the 3D structure of the p53 protein or impair its ability to bind DNA, leading to a loss-of-function [11].

*TP53* mutations are identified in nearly 30% of all breast cancers [12,13]. Numerous preclinical and clinical studies have explored the clinical significance of *TP53* mutations in breast cancer, with most associating them with poor prognosis [14,15,16,17]. However, studies that challenging the conventional understanding of the clinical relevance of *TP53* mutations have also been published. Ostrowski et al. found that while tumors with p53 expression exhibited more aggressive clinicopathological features, there was no significant difference in survival outcomes compared to tumors without p53 expression [18]. In a study by Shiao et al., which evaluated the association between p53 gene alterations and survival in patients with *TP53* mutations, differences in p53 gene alteration patterns were observed between Black and White patients. Among Black patients, *TP53*-mutated breast cancer was associated with poorer outcomes, whereas no such correlation was found in White patients [19]. Additionally, a meta-analysis including 26 studies with 3476 patients reported that patients with *TP53* mutations had a better response to neoadjuvant chemotherapy [20]. However, studies investigating the association between *TP53* mutations and breast cancer have generally been limited by small patient cohorts and prone to selection bias due to the varying prevalence of *TP53* mutations across molecular subtypes. Moreover, the lack of standardization in *TP53* mutation testing methods and treatment protocols, such as chemotherapy regimens, complicates the interpretation of findings. As a result, the clinical relevance of *TP53* mutations in breast cancer remains controversial.

As breast cancer treatment becomes increasingly personalized, there is a growing need not only to access the presence of *TP53* mutations but also to adopt a molecular approach to better understand these mutations. In breast tumors, gene sequencing revealed that missense mutations were dominant, accounting for about 80%, while other mutations, such as nonsense and frameshift mutations, made up approximately 20%. Additionally, the mutational events also differed from those observed in other cancers [21,22]. Moreover, *TP53* mutations in breast cancer act through various mechanisms, including impairing DNA damage repair, promoting cancer stemness, and enhancing inflammatory responses, each of which may require different therapeutic strategies [23,24,25]. Therefore, more in-depth research on the specific types and locations of *TP53* mutations is urgently needed. However, research in this area remains limited.

In light of these considerations, we investigated the association between *TP53* mutations and prognosis in breast cancer patients using long-term follow-up data. Additionally, we explored the clinical relevance of the characteristics of *TP53* mutations among patients harboring these mutations.

## 2. Materials and Methods

### 2.1. Data Collection

We retrospectively identified patients diagnosed with breast cancer who underwent *TP53* mutation testing at Gangnam Severance Hospital from January 2007 to December 2015. Clinicopathological data were collected from electronic medical records including age at diagnosis, histologic subtype, histologic grade, estrogen receptor (ER) and progesterone receptor (PR) status, human epidermal growth factor receptor 2 (HER2) status, lymphovascular invasion (LVI), Ki67 index, T stage, N stage, and implementation of (neo)adjuvant chemotherapy. We also collected genetic information about *TP53* mutation status and characteristics of *TP53* mutation. Patients diagnosed with recurrent breast cancer and de novo metastatic breast cancer were excluded. We also excluded bilateral breast cancer to minimize bias from concomitant pathologies.

T stage and N stage were determined using surgical specimens according to the American Joint Committee on Cancer Guidelines (AJCC) (8th edition). Hormone receptor (HR), ER and PR status was determined from surgical specimen using immunohistochemistry (IHC). Positive for ER and PR were defined as those in which more than 1% of tumor nuclei in the sample were stained [26]. HER2 status was assessed following the recommendation of the 2013 American Society of Clinical Oncology (ASCO)/College of American Pathologist (CAP) [27]. Triple-negative breast cancer (TNBC) refers to tumors that are negative for ER and PR, and do not exhibit HER2 overexpression, as determined by IHC. In this study, we applied a 20% threshold, commonly used in luminal-like subtypes, to classify Ki67 status as high or low, establishing a broadly applicable standard across all breast cancer subtypes [28,29]. Neoadjuvant and adjuvant systemic therapies, including chemotherapy, radiotherapy and endocrine therapy, were administered in accordance with established guidelines based on the age at diagnosis, molecular subtype, and axillary lymph node status.

### 2.2. Mutational Analysis of TP53 Gene

Mutational analysis of exons 5-9 of the *TP53* gene was performed using polymerase chain reaction—denaturing high performance liquid chromatography (PCR-DHPLC) and direct sequencing. Approximately 1 mg of samples from either biopsies or surgical specimens, freshly frozen of paraffin-embedded, were cut into pieces, and DNA was extracted using the Easy-DNA^TM^ kit (Invitrogen, Carlsbad, CA, USA) with 100 ng/µL of DNA used for each PCR reaction, where each PCR was performed in a 20 µL reaction mixture containing 100 ng of DNA, 20 µM of forward and reverse primers, 2 µL of Taq buffer (10×), 2.5 mM of deoxyribonucleotide triphosphates (dNTPs), 2.5 mM of MgCl_2_, and 0.7 U of Taq DNA polymerase, under conditions of 95 °C for 5 min, followed by 50 cycles of 94 °C for 10 s, 62 °C for 10 s, 72 °C for 15 s, and a final extension at 72 °C for 5 min in a DNA terminal cycler (Perkin-Elmer, GeneAmp PCR System 2400, Waltham, MA, USA), after which the PCR products were kept at 4 °C until further analysis, initially screened for mutations using DHPLC (WAVE; Transgenomic, Omaga, NE, USA), followed by sequence analysis if heteroduplex formation was detected, with DHPLC performed by mixing 20 µL of each exon PCR product with an equal amount of the corresponding wild-type PCR product, incubating at 95 °C for 5 min, and then at room temperature, and separating heteroduplex and homoduplex strands using triethylammonium acetate (TEAA) absorbed into the surface of the DNASep Cartridge (Transgenomic, USA) through an association with the negatively charged phosphate backbone of DNA, with elution using acetonitrile (ACN), in a gradient solution of buffer A (0.1 M TEAA solution, pH 7.0) and buffer B (0.1 M TEAA and 25% ACN, pH 7.0), with buffer C (8% ACN (syringe washing solution)) and buffer D (75% ACN (DNASep Cartridge Ultra-Clean and Storage Solution)) used for cleansing, while the stationary phase involved the DNASep Cartridge (Transgenomic, USA) column in an alkylated nonporous poly(styrene-divinylbenzene) form, washed with buffer D at 0.9 mL/min for 60 min, with the detection of separated DNA checked for purity by injecting 0.5 µL of the non-denatured specimen into the column at 0.9 mL/min at 50 °C, with the temperature elevated to 63 °C and the eluted DNA detected using an ultraviolet light detector at 260 nm, with analysis showing heteroduplexes eluted more rapidly than homoduplexes and appearing as separate forms in the chromatogram, and the DHPLC device operated per the manufacturer’s instruction, with denatured PCR products at 95 °C for 5 min, annealed at 55 °C for about 40 min, and monitored as a chromatogram, where heterogenous molecules typically displayed an additional peak compared to homozygous molecules, which had only one peak, and sequence analysis was performed using commercial reagents and an automated sequencer (ABI Prism BigDye Terminator v3.1 cycles sequencing kit and ABI 310 Genetic Analyzer; Applied Biosystems, Foster City, CA, USA), with both forward and reverse sequenced to confirm nucleotide alterations.

### 2.3. Definition of TP53 Mutation Characteristics and Oncologic Outcomes

In this study, we classified cases with mutations identified in exons 5-9 through DNA sequencing, as previously described [30,31], into the *TP53*-mutated group and cases with no mutations detected into the *TP53* wild-type group. To validate the clinical relevance of the characteristics of *TP53* mutation, we subcategorized the *TP53*-mutated group into some categories. Since most *TP53* mutations are missense mutations and are predominantly found in the DBD, we performed subgroup analyses by subdividing the *TP53*-mutated group into missense mutation vs. other mutations and DBD vs. other locations. Additionally, we distinguished and analyzed cases with missense hotspot mutations (missense mutations situated at codon 175, 220, 245, 248, 273, and 282) separately from other cases. The characteristics of *TP53* mutations within the *TP53*-mutated group are visualized in Figure 1.

Recurrence-free survival (RFS) was defined as the time from treatment of breast cancer (surgery or neoadjuvant chemotherapy) to relapse or death from any cause. Tumor recurrence occurring in the parenchyma of the ipsilateral breast affected by the primary cancer was defined as local recurrence (LR), and metastasis to the ipsilateral axillary lymph node, internal mammary node, and supraclavicular node were classified as regional recurrence (RR). Metachronous breast cancer (recurrence affecting the contralateral breast diagnosed after 1 year from the first cancer diagnosis [32]) was also defined as regional recurrence in this study. Metastasis to all other organs was defined as distant metastasis (DM). Overall survival (OS) was defined as the time from the treatment to death from any cause.

### 2.4. Statistical Analysis

We utilized the chi-square test or Fisher’s exact test to compare the proportion of de-mographic and clinicopathological variables between the two groups based on *TP53* mutation status. Comparisons among *TP53*-mutated subgroups, based on characteristics of *TP53* mutation including mutation types and locations, were also conducted. Oncologic outcomes between the two groups, classified according to *TP53* mutation status and characteristics, were compared using a stratified log-rank test at a two-sided significance level of 0.05. A stratified Cox regression analysis was performed to estimate hazard ratio (HR) and 95% confidence intervals (CIs) for oncologic outcome. To estimate the HR of each clinicopathological variable and *TP53* mutation status for RFS and OS, we performed Cox proportional hazard model. Multivariable Cox analyses were performed using all variables with *p*-value (*p*) ≤ 0.05. Statistical significance was set as *p* ≤ 0.05. All data analysis was conducted with SPSS software version 26.0 (SPSS Inc., Chicago, IL, USA) and GraphPad Prism software version 10.0 (GraphPad software Inc., Boston, MA, USA).

## 3. Results

### 3.1. Baseline Patient Characteristics

Between January 2007 and December 2015, 650 patients underwent *TP53* mutation testing using preoperative biopsies or surgical specimens at Gangnam Severance Hospital. Among these, there were 172 patients (26.5%) who detected *TP53* mutations. Of the patients with *TP53* mutations, 34 (19.8%) had missense hotspot mutations (Figure 2).

Table 1 presents the demographic and clinicopathological characteristics of patients according to *TP53* mutation status. The median age in both groups was 52 years. Compared to the *TP53* wild-type group, the *TP53*-mutated group had a higher proportion of ductal-type breast cancer (86.0% vs. 76.6%; *p* = 0.016), more frequent histologic grade III tumors (61.6% vs. 28.9%, *p* < 0.001), an increased rate of LVI (34.5% vs. 17.4%, *p* < 0.001), and a higher Ki67 index (73.8% vs. 31.4%, *p* < 0.001). Additionally, the *TP53*-mutated group had a higher incidence of HR-negative tumor (64.7% vs. 35.9%, *p* < 0.001) and a greater frequency of HER2-positive tumors (44.2% vs. 26.8%, *p* < 0.001). When categorized by molecular subtype, the *TP53*-mutated group exhibited a lower proportion of HR-positive/HER2-negative tumors (18.0% vs. 50.9%) and higher proportions of HER2-positive (44.2% vs. 29.4%) and triple-negative tumors (37.8% vs. 19.7%) compared to the *TP53* wild-type group (*p* < 0.001).

After excluding patients who received neoadjuvant chemotherapy, the distribution of T stage in the *TP53* wild-type group was 54.6% (253/463) for T1, 42.1% (195/463) for T2, and 3.2% (15/463) for T3-4. In the *TP53*-mutated group, the distribution was 42.9% (69/161) for T1, 53.4% (86/161) for T2, and 3.7% (6/161) for T3-4. There was no significant difference in the proportion of each N stage between the two groups (*p* = 0.922). In both groups, regardless of *TP53* mutation status, the mastectomy rate was higher than the breast-conserving surgery (BCS) rate; however, the mastectomy rate was lower in the *TP53*-mutated group compared to the *TP53* wild-type group (57.0% vs. 66.7%; *p* = 0.022). The majority of patients in both groups underwent sentinel lymph node biopsy (SLNB) alone, and while the axillary lymph node dissection (ALND) rate was higher in the *TP53*-mutated group, this difference was not statistically significant (18.0% vs. 12.6%; *p* = 0.152).

As previously mentioned, all patients included in the study received established standard treatment based on a comprehensive evaluation of their age at diagnosis, molecular subtype, and nodal metastasis. Patients in the *TP53*-mutated group were more likely to receive adjuvant chemotherapy (87.0% vs. 67.8%, *p* < 0.001) and post-operative radiotherapy (55.8% vs. 43.8%; *p* = 0.007) compared to those in the *TP53* wild-type group. In HER2-positive subtype, a total of 29 patients (14.2%, 29/204) did not receive HER2-targeted therapy due to advanced age or comorbidities; however, the difference in the proportion of these patients between the *TP53*-mutated and *TP53* wild-type groups was not statistically significant (12.0% in the *TP53*-mutated group vs. 17.7% in the *TP53* wild-type group; *p* = 0.289) (data not shown).

At the time of data cut-off of this study, the median follow-up period was 86.2 months [IQR, 60.3–111.8] in the *TP53*-mutated group and 97.4 months [IQR, 63.6–134.4] in the *TP53* wild-type group.

### 3.2. Oncologic Outcomes According to TP53 Mutation Status

With an extended follow-up period, we assessed 5-year and 10-year oncologic out-comes by using Kaplan-Meier analysis and Cox regression analysis. The RFS rates at 5-year were 88.1% (95% CIs, 84.1–91.1) in the *TP53* mutated-group, 93.7% (95% CIs, 91.0–95.7) in the *TP53* wild-type group, and the 10-year RFS rates were 83.5% (95% CIs, 76.2–88.8) in the *TP53*-mutated group, 86.6% (95% CIs, 80.2–91.1) in the *TP53* wild-type group, showing a statistically significant difference between the two groups (HR, 1.67; 95% CIs, 1.06–2.64; *p* = 0.026; Figure 3A).

The OS rates at 5-year were 89.8% (95% CIs, 83.8–93.6) in the *TP53*-mutated group and 95.3% (95% CIs, 92.8–97.0) in the *TP53* wild-type group, while the 10-year OS rates were 88.1% (95% CIs, 81.7–92.4) in the *TP53*-mutated group and 91.0% (95% CIs, 87.3–93.6) in the *TP53* wild-type group, indicating that the *TP53*-mutated group had a worse prognosis compared to the *TP53* wild-type group (HR, 3.02; 95% CIs, 1.43–6.70; *p* = 0.003; Figure 3B). However, when recurrence events were analyzed by sites, there were no differences between the two groups in terms of local recurrence-free survival (LRFS), regional recurrence-free survival (RRFS), and distant metastasis-free survival (DMFS) (Appendix A).

We utilized a Cox regression model to explore predictive factors for RFS and OS. Univariable analysis showed that *TP53* mutation was significantly associated with a shorter period of RFS (HR, 1.669; 95% CIs, 1.058–2.635; *p* = 0.028; Table 2) and OS (HR, 3.092; 95% CIs, 1.427–6.698; *p* = 0.004; Table 3). In multivariable analysis, which included all predictors with a *p* ≤ 0.05 from the univariable Cox analysis, *TP53* mutation remained an independent predictor of worse RFS (HR, 1.29; 95% CIs, 1.008–1.832; *p* = 0.046; Table 2) and OS (HR, 2.488; 95% CIs, 1.407–3.788; *p* = 0.044; Table 3). Additionally, the multivariable Cox analysis indicated that the presence of LVI and a high Ki67 index were significantly associated with worse RFS (Table 2), and the presence of LVI was also an independent predictor of worse OS (Table 3). In the univariable analysis, large tumor size (more than 2 cm) was identified as a factor associated with worse RFS and OS, but it was not statistically significant in the multivariable analysis. In addition, factors such as young age at diagnosis, high histologic grade, HR and HER2 positivity, nodal involvement, breast preservation during surgery, and the use of HER2-targeted therapy were not significantly associated with survival outcomes in our study.

### 3.3. Subgroup Analysis Based on Mutation Types Within the TP53-Mutated Group

Since most *TP53* mutations are known to be missense mutations, we conducted a subgroup analysis to examine potential differences in oncologic outcomes between missense mutation and other mutation types. Among the 172 cases with confirmed *TP53* mutations, 96 (55.8%) were missense mutations, and 76 (44.2%) were other types of mutations.

After excluding patients who received neoadjuvant chemotherapy, the missense mutation subgroup had a higher proportion of tumors ≤ 2 cm, and a lower proportion of tumors > 2 cm compared to the other mutations subgroup (T1 tumors; 50.6% in the missense mutation subgroup vs. 33.3% in the other mutations subgroup; *p* = 0.026). Consequently, patients in the missense mutation subgroup underwent BCS more frequently (53.1% vs. 30.3%; *p* = 0.003) and were more likely to receive post-operative radiotherapy (63.5% vs. 46.1%; *p* = 0.022) than those in the other mutations subgroup. However, no differences were observed between the two groups in the proportion of other clinicopathologic variables with surgery and treatment implementation. Detailed information is presented in Appendix A.

With a median follow-up period of 86.1 months (IQR, 54.1–110.8), there were no significant differences in RFS and OS between the two groups. The 5-year RFS rates were 89.9% (95% CIs, 81.4–94.6) in the missense mutation group and 82.3% (95% CIs, 70.8–89.5) in the other mutations group, whereas the rates of RFS at 10 years were 86.3% (95% CIs, 76.3–92.3) in the missense mutation group and 79.6% (95% CIs, 67.0–87.8) in the other mutations group (HR, 0.63; 95% CIs, 0.30–1.32; *p* = 0.217; Figure 4A). The 5-year OS rates were 93.9% (95% CIs, 84.5–97.7) in the missense hotspot mutation group and 93.2% (95% CIs, 85.4–96.9) in the other mutation group, while the 10-year OS rates were 88.0% (95% CIs, 76.3–94.2) in the missense mutation group and 93.2% (95% CIs, 85.4–96.9) in the other mutations group (HR, 1.63; 95% CIs, 0.55–4.84; *p* = 0.378; Figure 4B). Additionally, LRFS, RRFS, and DMFS did not differ significantly between the two groups (Appendix A).

### 3.4. Subgroup Analysis Based on Locations of Mutation Within the TP53-Mutated Group

Next, focusing on the patient with *TP53* mutation, we conducted a subgroup analysis to investigate oncologic outcomes based on the locations of *TP53* mutations. First, we classified the location of *TP53* mutations into the DBD and other locations. In total, there were 151 cases (87.8%) in the DBD subgroup and 21 cases (12.2%) in the other locations subgroup.

Compared to the other locations subgroup, the DBD subgroup had a lower proportion of HR-positive tumors (32.0% vs. 60.0%; *p* = 0.014). When tumors were classified by molecular subtype, the DBD subgroup exhibited a lower proportion of HR-positive/HER2-negative and HER2-positive tumors, and a higher proportion of triple-negative tumors (HR-positive/HER2-negative; 16.0% vs. 30.0%, HER2-positive; 42.0% vs. 60.0%, triple-negative; 42.0% vs. 10.0%; *p* = 0.018). However, there were no differences between the two groups in the proportion of the other collected variables (Appendix A).

As with *TP53* mutation type, there were no differences in oncologic outcomes between the two groups based on mutation locations. The 5-year RFS rates were 86.8% (95% CIs, 79.9–91.5) in the DBD group and 85.2% (95% CIs, 60.6–95.0) in the other locations group; and the 10-year RFS rates were 83.2% (95% CIs, 75.2–88.9) in the DBD group and 85.2% (95% CIs, 60.6–95.0) in the other locations group (HR, 0.79; 95% CIs, 0.24–2.63; *p* = 0.378; Figure 5A). The OS rates at 5 years were 93.2% (95% CIs, 87.4–96.4) in the DBD group and 95.2% (95% CIs, 70.7–99.3) in the other locations group, whereas the 10-year OS rates were 91.4% (95% CIs, 84.9–95.2) in the DBD group and 88.4% (95% CIs, 60.3–97.1) in the other locations group (HR, 0.24; 95% CIs, 0.27–5.58; *p* = 0.781; Figure 5B). LRFS, RRFS, and DMFS did not differ significantly between the two groups (Appendix A).

### 3.5. Subgroup Analysis Based on the Presence of Missense Hotspot Mutations Within the TP53-Mutated Group

Lastly, we analyzed oncologic outcomes by distinguishing between cases with missense hotspot domains and those without within the patients with *TP53* mutation. The majority of mutations identified at hotspot codons were missense mutations (39/44, 88.6%). However, no statistically significant differences in patients’ characteristics were observed between the two groups (Appendix A).

The median follow-up period was 95.1 months (IQR, 81.9–98.8) in the missense hotspot mutations group and 83.7 months (IQR, 75.6–89.3) in the other mutations group. At 5 years, the RFS rates were 100% in the missense hotspot mutations group and 83.4% (95% CIs, 75.6–88.8) in the other mutations group. The Kaplan-Meier estimates of 10-year RFS rates were 100% in the missense hotspot mutations group and 79.6% (95% CIs, 70.8–86.0) in the other mutations group, indicating that the missense hotspot mutations group had a better oncologic outcome than the other mutations group (HR, 0.15; 95% CIs, 0.06–0.39; *p* = 0.033; Figure 6A). However, there was no significant difference in OS between the two groups (HR, 0.70; 95% CIs, 0.18–2.66; *p* = 0.636; Figure 6B). Additionally, the RFS rates stratified by recurrence sites were not significantly different between the two groups (Appendix A).

### 3.6. Clinical Relevance of TP53 Within Molecular Subtypes of Breast Cancer

As part of an exploratory analysis, we examined the clinical relevance of *TP53* mutations within specific molecular subtypes. After excluding 72 patients for whom IHC-based HR status was unavailable, there were 239 patients (41.3%) with HR-positive/HER2-negative (HR+/HER2-) breast cancer, 188 patients (32.5%) with HER2-positive breast cancer, and 151 patients (26.1%) with TNBC. The proportion of patients with confirmed *TP53* mutations in each subtype were 30 patients (12.6%) in the HR+/HER2- subtype, 75 patients (39.9%) in the HER2-positive subtype, and 65 patients (43.0%) in the TNBC group. When comparing survival outcomes based on *TP53* mutation status within each subtype using the Kaplan-Meier estimated model, there were no differences in RFS or OS in HR+/HER2- (Figure 7A,B) and HER2-positive breast cancer (Figure 7C,D). However, in TNBC, patients with *TP53* mutation had worse RFS compared to those with *TP53* wild-type (HR, 2.13; 95% CIs, 1.01–4.50; *p* = 0.046; Figure 7E). Nevertheless, there was no statistical difference in OS according to *TP53* mutation status in TNBC (HR, 1.83; 95% CIs, 0.58–5.73; *p* = 0.295; Figure 7F). In addition, when comparing survival outcomes based on HER2 overexpression status within the *TP53*-mutated group, no statistically significant differences were observed (Appendix A).

## 4. Discussion

In this retrospective cohort study, we assessed the clinical relevance of *TP53* mutations in breast cancer patients, including all subtypes and treatments, and conducted subgroup analyses based on the characteristics of *TP53* mutations within the *TP53*-mutated group. *TP53* mutations were more frequent in breast cancer with more aggressive clinicopathological variables, such as large tumors, tumors with LVI or high histologic grade, and overexpression of HER2. Patients with *TP53* mutations had shorter RFS and OS compared to patients with *TP53* wild-type tumors. However, within the *TP53*-mutated group, the oncologic outcomes did not significantly differ between subgroups based on the characteristics of the *TP53* mutations. Missense mutation, mutation situated on DBD, and even missense mutation situated in hotspots, which are all well-known dominant characteristics of *TP53* mutation, did not have clinical relevance compared to other types or locations of *TP53* mutations. Although patients with missense hotspot mutations in the *TP53*-mutated group had a longer RFS period compared to other patients, there was no difference in OS rate. Therefore, the prognostic impact of missense hotspot mutations of *TP53* gene remains questionable.

Although *TP53* mutations are found in approximately 30% of all breast cancers [13], the proportion of these mutations varies by tumor subtypes. Furthermore, due to the differing mechanism of p53 protein among tumor subtypes and treatments, most studies on the clinical relevance of *TP53* mutations in breast cancer have been conducted within specific subtypes or treatments. Given that p53 regulates cell response to DNA damage, there have been several studies investigating the role of *TP53* mutations in patients undergoing chemotherapy or radiation, which induces tumor cell damage. Early preclinical trials indicated that p53 plays a role in regulating apoptosis or cell cycle arrest following cell damage such as radiation or systemic anticancer treatments [33,34,35,36]. Subsequent studies have shown that breast cancer patients with *TP53* mutations often have higher pathologic complete response (pCR) rates following neoadjuvant chemotherapy [37,38,39,40]. Otherwise, there were studies showing neutral and negative results regarding the association between *TP53* mutations and pCR rates following neoadjuvant chemotherapy [41,42,43]. Most of previous studies had small sample sizes and used different chemotherapy regimens and methods for detecting *TP53* mutations, making it challenging to define the clinical relevance of *TP53* mutations. Recently, a meta-analysis of 26 studies involving 3476 breast cancer patients who underwent neoadjuvant chemotherapy found that those with *TP53* mutations had a higher pCR rate [20]. However, even though this study confirmed the clinical relevance of *TP53* mutations through a sizable cohort, it also had the limitation of inconsistent *TP53* mutation detecting methods across the included studies. Additionally, most cases receiving neoadjuvant chemotherapy were HER2-positive breast cancer or TNBC. Therefore, it is difficult to consider these studies as having a balanced representation of all breast tumor subtypes.

ER-positive breast cancers account for about 70% of all breast cancers, making them the most prevalent subtype. In ER-positive breast tumors, the frequency of *TP53* mutations is lower than in other subtypes [40]; however, when these mutations are present, they are associated with a poor prognosis. Many studies presented that *TP53* mutations could lead to alterations in the p53 protein, potentially causing endocrine resistance [44,45,46]. However, the relationship between *TP53* mutations and survival outcomes in patients receiving only hormone therapy has been controversial [40,45,47]. This is due to several factors such as the small sample size, the detection of *TP53* mutations primarily through IHC, and the lack of information of additional treatments beyond hormone therapy. In a meta-analysis examining the clinical relevance of *TP53* mutations in patients receiving only hormone therapy, it was found that patients with *TP53* mutations had worse overall survival compared to those without *TP53* mutations [48]. Although a different dataset with varying *TP53* mutation detecting methods was utilized, we previously identified an association between *TP53* mutations and high 21-gene recurrence score in ER+HER2- breast tumors [49]. This finding aligns with prior research indicating that *TP53* mutations are associated with endocrine resistance in ER-positive breast tumors. Compared to ER-positive breast cancer, ER-negative breast cancer accounts for a smaller proportion of all breast tumors; however, the frequency of *TP53* mutations is higher in ER-negative breast cancer. *TP53* mutation rates are higher in HER2-positive and TNBC (also referred to as basal-like subtype) compared to luminal-type breast cancer, which are predominantly ER-positive tumors [13,50,51,52,53,54,55,56]. Some studies indicated that the presence of *TP53* mutations is associated with poor prognosis and might confer resistance to chemotherapy in HER2-positive breast cancer and TNBC [54,57,58,59]. However, some studies showed no difference in oncologic outcomes based on *TP53* mutation status in ER-negative tumors [60,61,62,63,64], or even suggested that *TP53* mutations are associated with better prognosis [39,65,66]. This trend has become more pronounced in recent studies as chemotherapy regimens have continuously evolved and the clinical use of new drugs, such as dual HER2 blockade and immune checkpoint inhibitors, has increased. Consequently, determining the clinical significance of *TP53* mutations in ER-negative breast cancer has become even more challenging. Given these circumstances, conducting studies to determine the clinical relevance of *TP53* mutations across all subtypes and treatments involves many hurdles and interpreting the results is also challenging.

Therefore, the strength of our study is its ability to assess long-term oncologic outcomes using a large cohort that encompasses all breast cancer subtypes and treatments. Excluding 42 patients whose hormone receptor status was not clearly identified, the data for this study included 253 HR-positive, HER2-negative tumors (41.6%), 204 HER2-positive cases (33.6%), and 172 TNBC (28.3%), which means the distribution of tumor subtypes in the collected data was well-balanced. To date, few studies have investigated the clinical relevance of *TP53* mutations using cohorts that included all breast cancer subtypes and treatments. In most of these studies, patients with *TP53* mutations were found to have worse survival compared to the *TP53* wild-type group [67,68,69,70]. However, these studies had limitations such as small sample size, lack of follow-up data, and inconsistent treatments even within the same subtype. This study, leveraging a large cohort from a single center, ensured consistent treatments according to tumor subtypes and stage, thereby minimizing bias from the data.

Another notable strength of our study is its focus on an Asian population, unlike most previous research on the link between *TP53* mutations and poor prognosis in breast cancer, which has primarily included individuals of American and European descent. Large-scale retrospective cohort studies evaluating the clinical relevance of *TP53* mutations in Asian breast cancer patients are still limited. Thus, our findings provide a unique opportunity to contribute evidence that could support cross-ethnic comparisons and validate the clinical implications of *TP53* mutations in breast cancer. In addition, by collecting data from patients who underwent *TP53* mutation testing between 2007 and 2015, we were able to secure comprehensive long-term follow-up data.

Furthermore, few studies have examined surgical outcomes based on the location and type of *TP53* mutations in patients with confirmed mutations, underscoring the significance of our investigation. Given that mutations causing loss of DNA-binding can critically affect the biological activity of p53 [71], there is increasing interest in understanding the characteristics of different *TP53* mutations. However, the clinical relevance of specific mutation types and locations remains underexplored. For instance, an analysis of the METABRIC cohort found that tumors harboring missense mutations in DNA-binding motifs (DBM) had a higher risk of breast cancer-specific mortality compared to tumors with non-missense mutations or missense mutations outside the DBM, though this difference did not reach statistical significance [64]. Similarly, a study in China demonstrated that metastatic breast cancer patients with *TP53* mutations outside the DBD experienced poorer disease-free survival and OS compared to those with *TP53* wild-type, with particularly poor outcomes observed in patients with non-missense mutations located within the DBD [72]. Pal, et al. assessed the 10 most common *TP53* missense mutations using MCF10A cell lines for preclinical investigation and found that mutations such as R248W, R273C, R248Q, and Y220C were associated with the most aggressive tumor phenotypes [73]. Børresen, et al. reported that *TP53* mutations in the zinc-binding domain were associated with worse prognosis compared to mutations outside this domain [74]. Meanwhile, Kucera, et al. observed no significant difference in survival outcomes between cases with mutations in the L2/3 domain and those without such mutations [75]. Data from the BIG 02-98 phase III trial also indicated that only truncating mutations were predictive of increased recurrence risk, while missense mutations showed no significant association [76]. Despite these findings, many studies have faced challenges in achieving statistical significance due to limited sample size, diverse patient populations, varying methods for analyzing *TP53* mutations, and differences in study endpoints. In our study, we investigated the clinical relevance of several characteristics within the *TP53*-mutated group, including missense mutations and mutations located in the DBD, but did not find statistically significant results. Nonetheless, considering the limited research focused on the clinical implications of *TP53* mutations in early breast cancer among Asian populations, we believe that our findings add valuable evidence to the existing literature and can help guide future research efforts.

Although our study allowed us to assess the clinical relevance of *TP53* mutations and their characteristics within a large cohort encompassing all tumor subtypes and treatments, it still had inherent limitations. The first limitation is the sensitivity of *TP53* mutations. In our study, we identified *TP53* mutations in exons 5-9 using PCR-DHPLC and direct sequencing. Although most *TP53* mutations occur within exon 5-9, this approach might lead to false-negative results for mutations occurring in other regions, particularly in exons 2-4 and 10-11 [77,78]. In addition, somatic mutations identified by PCR-DHPLC might not always be detectable by direct sequencing, because it has a threshold of detection of approximately 15–20% [79]. To overcome this limitation, NGS is now used for DNA sequencing in breast cancer [80]. However, since NGS was introduced at our institution in 2017, it was not applied for the patients retrospectively collected for this study. Another limitation of our study is the reliance on older data, which may have constrained our ability to control confounding variables effectively. Furthermore, although this study included a broad cohort covering all breast cancer subtypes, the overall number of patients and the subgroup sizes within the *TP53* mutation category were limited, representing a notable study limitation. In our Cox regression analysis assessing associations between clinicopathological features and survival outcomes, several established prognostic and predictive markers did not reach statistical significance, likely due to the sample size constraints, which may have reduced the power to detect meaningful associations. Lastly, since the data were collected 10 years ago, the treatment protocols at that time may differ significantly from those currently used in clinical practice. Most patients in this study did not receive neoadjuvant chemotherapy, and treatments such as CDK4/6 inhibitors, immune checkpoint inhibitors, and dual HER2 blockade including pertuzumab were rarely administered at that time. Nevertheless, based on this study, we expected that we could conduct further research addressing a prognostic influence on the characteristics of *TP53* mutations in breast cancer patients with advanced research methods and molecular studies.

## 5. Conclusions

Using consistently collected long-term follow-up data, we found that *TP53* mutations are associated with worse prognosis in breast tumors encompassing all subtypes and treatments. Additionally, within the *TP53*-mutated group, there were no significant differences in surgical outcomes based on the characteristics of *TP53* mutations such as mutation types and locations of mutation.

## Figures and Tables

**Figure 1 cancers-16-03899-f001:**
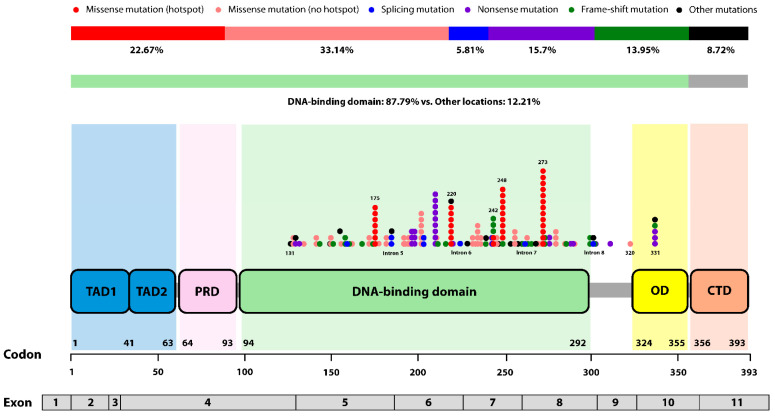
Characteristics of *TP53* mutations in patients within the *TP53*-mutated group. More than half of the identified *TP53* mutations were missense mutations, with the majority occurring in the DNA-binding domain (DBD). Each circle represents a codon where a *TP53* mutation occurred, with mutation type distinguishing by color. The number of circles indicate the total number of mutations occurring within specific codons. (Abbreviation, TAD; transactivation domain, PRD; proline-rich domain, OD; oligomerization domain, CTD; carboxy-terminal domain).

**Figure 2 cancers-16-03899-f002:**
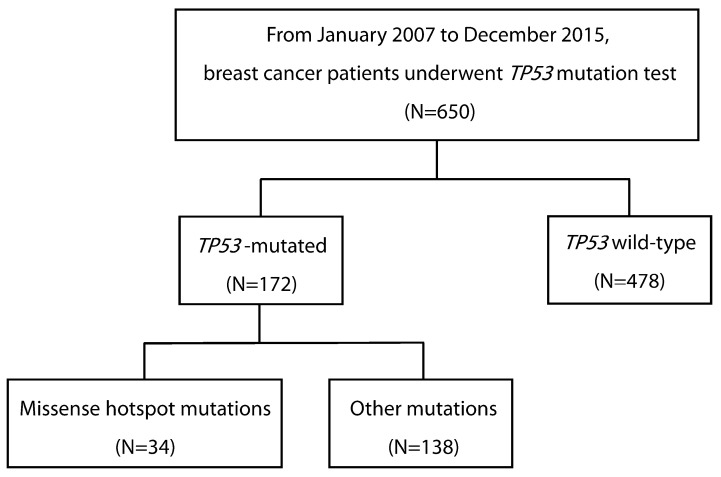
Consort diagram of this study.

**Figure 3 cancers-16-03899-f003:**
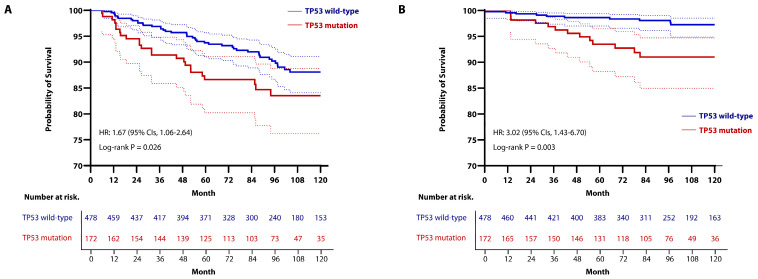
Kaplan-Meier curve for (**A**) RFS and (**B**) OS in patients stratified by *TP53* mutation status. (**A**) Stratified log-rank test and Cox regression analysis showed a significant different between the two groups (The 5-year RFS rates: 88.1% (95% CIs, 84.1–91.1) in the *TP53*-mutated group vs. 93.7% (95% CIs, 91.0–95.7) in the *TP53* wild-type group; the 10-year RFS rates; 83.5% (95% CIs, 76.2–88.8) in the *TP53*-mutated group vs. 86.6% (95% CIs, 80.2–91.1) in the *TP53* wild-type group) (HR, 1.67; 95% CIs 1.06–2.64; *p* = 0.026). (**B**) Stratified log-rank test and Cox regression analysis showed a significant difference between the two groups (The 5-year OS rate: 89.8% (95% CIs, 83.8–93.6) in the *TP53*-mutated group vs. 95.3% (95% CIs, 92.8–97.0) in the *TP53* wild-type group; the 10-year OS rate: 88.1% (95% CIs, 81.7–92.4) in the *TP53*-mutated group vs. 91.0% (95% CIs, 87.3–93.6) in the *TP53* wild-type group) (HR, 3.02; 95% CIs, 1.43–6.70; *p* = 0.003).

**Figure 4 cancers-16-03899-f004:**
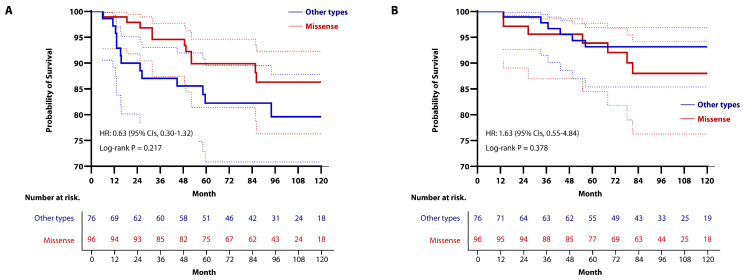
Kaplan-Meier curve for (**A**) RFS and (**B**) OS in the *TP53*-mutated group, stratified by type of mutation. To compare oncologic outcomes, we used stratified log-rank test and Cox regression analysis. (**A**) There was no significant difference between the two groups (the 5-year RFS rates: 89.9% (95% CIs, 81.4–94.6) in the missense mutation group vs. 82.3% (95% CIs, 70.8–89.5) in the other mutations group; the 10-year RFS rates: 86.3% (95% CIs, 76.3–92.3) in the missense mutation group vs. 79.6% (95% CIs, 67.0–87.8) in the other mutations group) (HR, 0.63; 95% CIs, 0.30–1.32; *p* = 0.217). (**B**) Similarly, there was no significant difference between the two groups (The 5-year OS rates: 93.9% (95% CIs, 84.5–97.7) in the missense mutation group vs. 93.2% (95% CIs, 85.4–96.9) in the other mutations group; the 10-year OS rates: 88.0% (95% CIs, 76.3–94.2) in the missense mutation group vs. 93.2% (95% CIs, 85.4–96.9) in the other mutations group) (HR, 1.63; 95% CIs, 0.55–4.84; *p* = 0.378).

**Figure 5 cancers-16-03899-f005:**
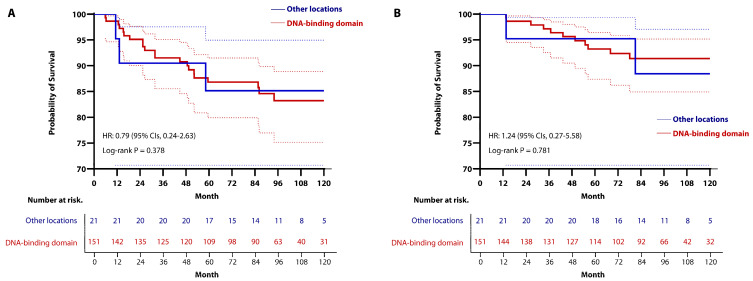
Kaplan-Meier curve for (**A**) RFS and (**B**) OS in the *TP53*-mutated group, stratified by mutation locations. To compare oncologic outcomes, we used stratified log-rank test and Cox regression analysis. (**A**) There was no significant difference between the two groups (the 5-year RFS rates: 86.8% (95% CIs 79.9–91.5) in the DBD group vs. 85.2% (95% CIs, 60.6–95.0) in the other locations group; the 10-year RFS rates: 83.2% (95% CIs, 75.2–88.9) in the DBD group vs. 85.2% (95% CIs, 60.6–95.0) in the other locations group) (HR, 0.79; 95% CIs, 0.24–2.63; *p* = 0.378). (**B**) There was also no significant difference between the two groups (the 5-year OS rates: 93.2% (95% CIs, 87.4–96.4) in the DBD group vs. 95.2% (95% CIs, 70.7–99.3) in the other locations group; the 10-year OS rates: 91.4% (95% CIs, 84.9–95.2) in the DBD group vs. 88.4% (95% CIs, 60.3–97.1) in the other locations group) (HR, 1.24; 95% CIs, 0.27–5.58; *p* = 0.781).

**Figure 6 cancers-16-03899-f006:**
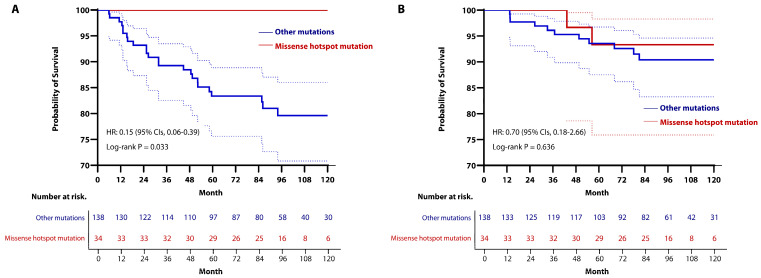
Kaplan-Meier curve for (**A**) RFS and (**B**) OS in patients with *TP53* mutation, stratified by the presence or absence of missense hotspot mutations. (**A**) By utilizing stratified log-rank test and Cox regression analysis, patients with missense hotspot mutations had a longer RFS period (the 5-year RFS rates: 100% in the missense hotspot mutations group vs. 83.4% (95% CIs, 75.6–88.8) in the other mutations group; the 10-year RFS rates: 100% in the missense hotspot mutations group vs. 79.6% (95% CIs, 70.8–86.0) in the other mutations group (HR, 0.15; 95% CIs, 0.06–0.39; *p* = 0.028). (**B**) Stratified log-rank test and Cox regression analysis showed that there was no significant difference between the two groups (the 5-year OS rates: 93.3% (95% CIs, 75.9–98.3) in the missense hotspot mutations group vs. 93.6% (95% CIs, 87.6–96.7) in the other mutations group; the 10-year OS rates; 93.3% (95% CIs, 75.9–98.3) in the missense hotspot mutations group vs. 90.4% (95% CIs, 83.3–94.6) in the other mutations group (HR, 0.70; 95% CIs, 0.18–2.66; *p* = 0.636).

**Figure 7 cancers-16-03899-f007:**
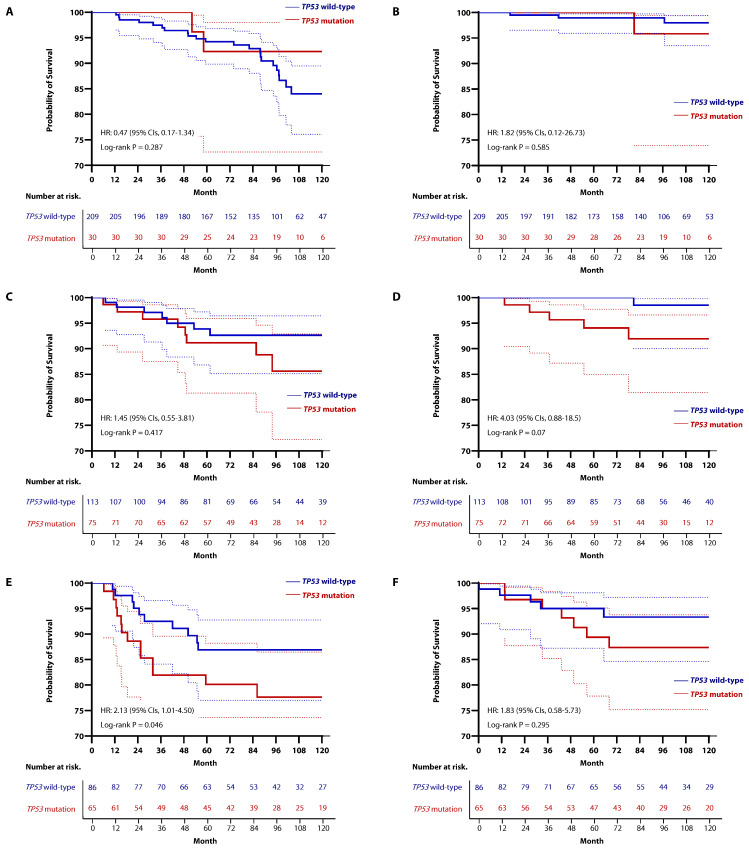
Kaplan-Meier curve for RFS and OS according to *TP53* mutation status in each subtype. In HR+/HER2-subtype, there was no difference in (**A**) RFS (HR, 0.47; 95% CIs, 0.17–1.34; *p* = 0.287) or (**B**) OS (HR, 1.82; 95% CIs; 0.12–26.73; *p* = 0.585) between *TP53*-muated group and *TP53* wild-type group. Similarly, in the HER2-positive subtype, no differences were observed in (**C**) RFS (HR, 1.45; 95% CIs; 0.55–3.81; *p* = 0.417) or (**D**) OS (HR, 4.03; 95% CIs; 0.88–18.5; *p* = 0.07) based on *TP53* mutation status. In the TNBC group, (**E**) *TP53*-mutated tumors showed worse RFS compared to the *TP53* wild-type group (HR, 2.13; 95% CIs; 1.01–4.50; *p* = 0.046). However, (**F**) although there was a trend toward worse OS in *TP53*-mutated tumors, it was not statistically significant (HR, 1.83; 95% CIs, 0.58–5.73; *p* = 0.295).

**Table 1 cancers-16-03899-t001:** Baseline patients’ characteristics according to *TP53* mutation status.

	*TP53*-Mutated (N = 172)	*TP53* Wild-Type (N = 478)	*p*-Value
Age, median [IQR]	52 [27–78]	52 [51–87]	0.284
Histologic subtype			0.016
Ductal	148 (86.0)	366 (76.6)	
Lobular	2 (1.2)	23 (4.8)	
Others and Mixed	22 (12.8)	89 (18.6)	
Histologic grade			<0.001
Grade III	106 (61.6)	138 (28.9)	
Grade I-II	66 (38.4)	340 (71.1)	
HR status ^#^			<0.001
Positive	60 (35.3)	261 (64.1)	
Negative	110 (64.7)	146 (35.9)	
HER2 status			<0.001
Positive	76 (44.2)	128 (26.8)	
Negative	96 (55.8)	350 (73.2)	
Molecular subtype ^#^			<0.001
HR-positive/HER2-negative	30 (17.6)	209 (51.2)	
HER2-positive	75 (44.1)	113 (22.7)	
Triple-negative	65 (38.2)	86 (21.1)	
LVI ^#^			<0.001
Positive	59 (34.5)	83 (17.4)	
Negative	112 (65.5)	395 (82.6)	
Ki67 index (cutoff 20%)			<0.001
High	127 (73.8)	150 (31.4)	
Low	45 (26.2)	328 (68.6)	
Neoadjuvant chemotherapy			0.062
Yes	11 (6.4)	15 (3.1)	
No	161 (93.6)	463 (96.9)	
T stage *			0.035
T1	69 (42.9)	253 (54.6)	
T2	86 (53.4)	195 (42.1)	
T3-4	6 (3.7)	15 (3.2)	
N stage *			0.826
N0	97 (62.2)	274 (60.8)	
N1	47 (30.1)	135 (29.9)	
N2-3	12 (7.7)	42 (9.3)	
Breast operation			0.022
BCS	74 (43.0)	159 (33.3)	
Mastectomy	98 (57.0)	319 (66.7)	
Axilla surgery			0.152
No approach	6 (3.5)	12 (2.5)	
SLNB	135 (78.5)	406 (84.9)	
ALND	31 (18.0)	60 (12.6)	
Adjuvant chemotherapy *			<0.001
Yes	140 (87.0)	314 (67.8)	
No	21 (13.0)	149 (32.2)	
Post-operative radiotherapy			0.007
Yes	96 (55.8)	210 (43.9)	
No	76 (44.2)	268 (56.1)	

^#^ Patients for whom accurate test values could not be confirmed were excluded. * Patients who received neoadjuvant chemotherapy or did not undergo surgery were excluded. Abbreviations, IQR; inter-quartile range, HR, hormone receptor, HER2; human epidermal growth factor receptor 2, LVI; lymphovascular invasion, BCS; breast-conserving surgery, SLNB; sentinel lymph node biopsy, ALND; axillary lymph node dissection.

**Table 2 cancers-16-03899-t002:** Univariable and multivariable analyses for RFS.

Variable	Univariable			Multivariable		
	HR	95% CIs	*p*-Value	HR	95% CIs	*p*-Value
Age ≤ 50 years (ref. > 50 years)	1.24	0.813–1.893	0.318			
*TP53* mutation (ref. *TP53* wild-type)	1.669	1.058–2.635	0.028	1.29	1.008–1.832	0.046
Histologic grade III (ref. HG I-II)	1.123	0.730–1.730	0.597			
HR positive (ref. HR negative) ^#^	1.234	0.781–1.950	0.367			
HER2 positive (ref. HER2 negative)	0.718	0.439–1.172	0.185			
LVI present (ref. LVI absent) ^#^	2.604	1.686–4.021	<0.001	2.366	1.495–3.747	<0.001
Ki67 high (≥20%) (ref. Ki67 < 20%)	1.829	1.198–2.790	0.005	1.607	1.030–2.506	0.037
Tumor > 2 cm (ref. Tumor ≤ 2 cm) *	1.743	1.115–2.724	0.015	1.355	0.843–2.178	0.209
Nodal involvement (ref. Node-negative) *	1.3	0.839–2.014	0.241			
BCS (ref. Mastectomy)	1.383	0.903–2.117	0.136			
HER2-targeted therapy (ref. no treatment)	0.774	0.455–1.316	0.343			

^#^ Patients without definite data were excluded. * Patients who underwent neoadjuvant chemotherapy were excluded. Abbreviations, HR, hazard ratio, CIs; confidence intervals, HR, hormone receptor, HER2; human epidermal growth factor receptor 2, LVI; lymphovascular invasion, BCS; breast-conserving surgery.

**Table 3 cancers-16-03899-t003:** Univariable and multivariable analyses for OS.

Variable	Univariable			Multivariable		
	HR	95% CIs	*p*-Value	HR	95% CIs	*p*-Value
Age ≤ 50 years (ref. > 50 years)	0.812	0.375–1.757	0.597			
*TP53* mutation (ref. *TP53* wild-type)	3.092	1.427–6.698	0.004	2.488	1.407–3.788	0.044
Histologic grade III (ref. HG I-II)	0.909	0.405–2.038	0.816			
HR positive (ref. HR negative) ^#^	0.549	0.244–1.238	0.148			
HER2 positive (ref. HER2 negative)	1.017	0.442–2.339	0.968			
LVI present (ref. LVI absent) ^#^	2.604	1.686–4.021	<0.001	2.366	1.495–3.747	<0.001
Ki67 high (≥ 20%) (ref. Ki67 < 20%)	2.419	1.096–5.340	0.029	2.35	0.966–5.717	0.06
Tumor > 2 cm (ref. Tumor ≤ 2 cm) *	2.781	1.079–7.170	0.034	2.061	0.765–5.551	0.153
Nodal involvement (ref. Node-negative) *	1.69	0.718–3.980	0.23			
BCS (ref. Mastectomy)	0.927	0.413–2.085	0.855			
HER2-targeted therapy (ref. no treatment)	0.914	0.367–2.276	0.846			

^#^ Patients without definite data were excluded. * Patients who underwent neoadjuvant chemotherapy were excluded. Abbreviations, OS; overall survival, HR, hazard ratio, CIs; confidence intervals, HR, hormone receptor, HER2; human epidermal growth factor receptor 2, LVI; lymphovascular invasion, BCS; breast-conserving surgery.

## Data Availability

The data that support the findings of this study contain clinical outcomes for which institutional review board (IRB) approval is required before analysis. Therefore, these data are not publicly available. The data will be provided to authorized researchers who have obtained IRB approval from their institution and Gangnam Severance Hospital, Yonsei University, Seoul, Korea. For data access requests, please contact the corresponding authors, namely Pf. J.J., email address: gsjjoon@yuhs.ac.

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
