# Peer review of "Clinical Relevance of TP53 Mutation and Its Characteristics in Breast Cancer with Long-Term Follow-Up Date"

_cancers, 2024, doi:10.3390/cancers16233899_

Round 1
Reviewer 1 Report
Comments and Suggestions for Authors
1. Does the extent of surgical treatment (mastectomy or sectoral resection) affect the prognosis? This data is not available in Table 1.
2. Is it not specified whether targeted therapy was performed in patients with HER2(+) status? If so, would it be more correct to check the effect of the P53 mutation separately in subgroups with different HER2 expression status? This factor is certainly taken into account in the multivariate analysis, but the factor of the presence of targeted therapy is not taken into account.
3. The presence of concomitant pathologies is not specified, this is also an important factor that should be taken into account during long-term observation.
Author Response
Thank you for your nice review and comments
Please see the attachment.

Reviewer 2 Report
Comments and Suggestions for Authors
Dear Authors,
I read with great interest your work. The article is well done and the thema is of high interest.
I would suggest you to introduce in the discussion a paragraph about clinical relevance of this mutation in comparison with other mutations which are used in the clinical practice; also relevance in terms of survival/therapy.
Yours,
Author Response
Thank you for your nice comment and insightful commment.
Please see the attachment.

Reviewer 3 Report
Comments and Suggestions for Authors
The manuscript presented is interesting, the authors evaluate the impact of p53 mutations on the clinical outcome of breast cancer patients. They have a large cohort and a long follow-up period, which is important. However, the manuscript could be improved with a few minor observations:
First, I think it is very important for you to include the acceptance number of the of your protocol by an ethics and research committee.
Clarify why you excluded patients with de novo metastases.
Why did you did you consider a high Ki67 value to be greater than 20%? Put reference.
It is understood that the detection of p53 mutations was performed by the same method, but it should be made explicit in the text.
Can you provide a list of missense mutations, hot spot mutations and others that were found?.
The authors mention that they sequenced exons 5 to 9 of p53, in which region of the protein do they fall or in which domain?.
It is necessary to provide the sequences of their oligos of forward and reverse primers.
Strangely, the authors found that factors such as young age at diagnosis, high histologic grade, HR and HER2 positivity, nodal involvement, breast preservation during surgery, and use of HER2-targeted therapy were not significantly associated with survival outcomes in their study. An explanation is needed.
Author Response
Thank you for your comprehensive comments and reviews.
Please see the attachment

Reviewer 4 Report
Comments and Suggestions for Authors
This study retrospectively included 650 breast cancer patients, including 172 with TP53 mutations, to evaluate the prognosis outcome of patients with TP53 mutations. The study design was straightforward and well-described, and the manuscript was well-written in general. I only have some minor suggestions.
Abstract, Methods: The location to recruit the patients need to be mentioned in the abstract.
Methods, line 81: Write the specific name of the institution instead of “our institution”.
Results, Table 2, 3: RFS and OS should be also included by the abbreviations in the footnote of table 2 and 3.
Discussion: A potential novelty of this study is that the participants were females of Asian ancestry, but this was not discussed in the manuscript. In the introduction (line 70, ref 14-17), it is better to summarize whether previous studies on TP53 and poor prognosis were dominantly conducted among European descendants. Then, some discussion about the females of Asian ancestry could be added to the manuscript.
Author Response
Thank you for your meticulous reviews
Please see the attachment.

Round 2
Reviewer 1 Report
Comments and Suggestions for Authors
The authors responded to the reviewer's comments in detail and substantially revised the manuscript. I have no further comments on the manuscript.
Author Response
Thank you for your valuable comments and insights.
Because of your wonderful advice, we could complete an excellent manuscript.
Thank you again for your help!
Reviewer 2 Report
Comments and Suggestions for Authors
well done! deserves to be published
Author Response

(The authors gave the same response as above.)
